# The Effects of Online Social Interactions on Life Satisfaction of Older Chinese Adults: New Insights Based on a Longitudinal Approach

**DOI:** 10.3390/healthcare10101964

**Published:** 2022-10-08

**Authors:** Dong Zhou, Yi Xu, Pengya Ai

**Affiliations:** 1Department of Cultural Industry and Management, Shanghai Jiao Tong University, Shanghai 200240, China; 2USC-SJTU Institute of Cultural and Creative Industry, Shanghai Jiao Tong University, Shanghai 200240, China; 3Wee Kim Wee School of Communication and Information, Nanyang Technological University, Singapore 637718, Singapore

**Keywords:** life satisfaction, Chinese older adults, online social interactions, social engagement, longitudinal study

## Abstract

Population aging and digitalization have become universal phenomena. Over the past two decades, digital inclusion has started to play a crucial role in supporting successful aging. Based on a nationally representative sample of around 5200 older adults in China over the period of 2014–2018, we explore the effects of online social interactions (OSIs) on the life satisfaction of older adults. We find that OSIs can improve the levels of life satisfaction of older Chinese adults. Estimates from fixed effect and cross-lagged structural equation models further suggest that OSIs work by increasing physical activities, healthy time allocation, interpersonal trust, and informal social engagement and reducing loneliness. We also find that OSIs narrow the social inequality in life satisfaction across groups from rural–urban areas and groups with different social statuses. Moreover, a comparison among different online engagements shows that not all online activities positively affect older adults’ life satisfaction. Different online activities have varying effects. Our results highlight that public digital interventions focusing on social functions can benefit the lives of older adults.

## 1. Introduction

Today, the world is facing two important and dramatic transitions: social and economic challenges related to the rapid aging population and advancement in digital technologies [1,2]. Globally, an estimated 728 million persons are 65 years or older (United Nations, 2020). Additionally, concerns have been raised over accelerating aging in China [2,3,4]. In 2021, residents aged 60 and above numbered 267.36 million, accounting for 18.9% of the national population, while those aged 65 and above numbered 201 million, accounting for 14.2%. Given the large population of older adults, their well-being is of great significance for enhancing social welfare. It is also one of the most pressing issues national policymakers must address.

The other essential transition is the advancement of digital technologies, which likewise pose substantial social and economic concerns. As of January 2021, global internet penetration now stands at 59.5%, and 4.66 billion people worldwide use the internet [5]. By the end of 2021, China had 1.032 billion internet users, accounting for 73% of the total population [6]. Notably, the proportion of older adult internet users also increased significantly (as shown in Figure A1 in Appendix A). As of December 2021, the number of internet users aged 60 and above reached 119 million in China, representing 11.5% of the total internet users and 43.2% of this age group. Over 85% of older internet users participate in social networking, suggesting that online social engagement has become an important part of the lives of many older adults. Thus, it is valuable to understand the impact of digital technologies on the subjective well-being of older adults. In the existing literature, life satisfaction as the most commonly used indicator of subjective well-being is largely interchangeable with subjective well-being, although they are not equivalent [7,8]. This paper explores how online social interactions affect the life satisfaction of older people in China.

Our study adds to the growing literature examining the impact of internet adoption on subjective well-being, particularly for older adults. To date, a large body of literature has identified benefits of internet use, including reduced loneliness [9,10,11] and a higher level of self-reported satisfaction as well as a lower chance of being isolated [12]. However, some investigations found an insignificant relationship between internet use and older adults’ subjective well-being [13,14]. Choi and DiNitto [15] even found that internet use among older adults was associated with higher anxiety and depressive symptoms. In the case of China, two studies examined the association between internet adoption and the subjective well-being of older adults. Xu and Huang [16] used cross-sectional data and found that internet use enhances happiness by reducing loneliness and increasing volunteering. Using the 2018 wave of CFPS, Lu and Kandilov [3] discovered that mobile internet use contributes to higher levels of subjective well-being in older adults.

Most of the inquiries acknowledged the positive association between subjective well-being and older adults but also called for more robust research into the possible causal relationship. One major concern is that most of the research used cross-sectional designs. The absence of longitudinal data makes causal inferences difficult. One close study is by Szabo, et al. [17], in which they evaluated the influences of different online activities of respondents in 2013 on their happiness in 2016. Similarly, Hartanto, et al. [18] employed a two-wave, cross-lagged design to evaluate the relationship between computer use and healthy, cognitive, and social benefits among middle-aged and older people. Both studies focused on older American adults. Strictly speaking, their longitudinal mediation analyses were cross-sectional studies with lagged structural equation modeling. Our study improves the majority of existing literature by providing a longitudinal study in China with individual fixed-effect models. In this panel data estimation method, we eliminate endogenous selection bias caused by individual unobserved factors. Moreover, previous research focused on the frequency of internet use or a broad measure of internet use rather than engagement in different online uses. Our study complements the existing research by comparing different online behaviors and investigating potential causal mechanisms.

## 2. Literature Review

As population aging is accelerating worldwide, academics and policymakers are increasingly interested in the factors influencing the SWB of older adults. Technological factors used to be ignored in the function of older adults’ well-being because their adoption of newborn technology largely lagged behind younger age groups. However, due to the prevalence of internet-based ICT and its significance in people’s lives, older adults have adopted the internet at a much faster pace than before. For example, in the United States, the internet use of older adults aged 65 and above grew to 75% in 2021 [19]. Correspondingly, research relating internet use to the SWB of older adults has recently begun to thrive.

Dating back to 2002, Chen and Persson surveyed 396 young and older adults in Northwest Ohio [20]. They found that older internet users were more positive than non-users concerning psychological well-being, unlike negative relations found for young adults. In 2007, Shapira et al. employed a quasi-experimental design and compared 22 trained older adults with 26 other untrained older adults in Israel [21]. They concluded that internet use could improve life satisfaction by changing interpersonal interactions, promoting cognitive functioning, and contributing to self-control. Mellor, et al. [22] conducted an internet training project in Australia and studied the impacts of internet use on SWB with 20 older adults. A total of 8 participants reported positive outcomes after 12 months, but the other 12 dropped the experiment after six months. Due to the limited sample size, the external validity of these experimental investigations is compromised.

As internet technology rapidly advances, household-level surveys have started to include questions on internet adoption, enabling researchers to study more comprehensively. Cotten, et al. [23,24] evaluated the impact of internet use on depression among the older, retired population in America. Both papers found that internet adoption can reduce depression. Moreover, Lelkes [12] used a European multi-country cross-sectional dataset with over 11,000 observations and found that internet use could reduce loneliness and enhance social relations, hence improving happiness among older adults. Quintana, et al. [25] also found that internet/email use is positively associated with the psychological well-being of adults aged 50 or above in the UK. Nakagomi, Shiba, Kawachi, Ide, Nagamine, Kondo, Hanazato and Kondo [7] studied older Japanese adults and found the frequency of internet use in 2016 was associated with better health outcomes and subjective well-being in 2019.

However, the causal relationship between internet use and older adults’ subjective well-being is still weak [26]. Some investigations reported no direct association between internet use and various mental health outcomes. For example, analyzing the National Health and Aging Trends Study, Elliot, Mooney, Douthit and Lynch [13] found that internet use was unrelated to depression symptoms or SWB. Wong, Yeung, Ho, Tse and Lam [14] also found no significant relationship between the frequency of internet use and mental health among older Chinese adults. Huang [27] and Nie, et al. [28] reported negative impacts of internet use on the psychological well-being of older adults, although the analyses focused on the general population. Choi and DiNitto [15] even found that internet use among older adults was associated with higher anxiety and depression.

These inconsistencies may be attributable to diverse personal characteristics, cultures, and beliefs [29]. Second, they might be related to cross-sectional designs, which cannot elucidate causality. Third, most measures of internet adoption in the existing literature are frequency of internet use or a general indicator for internet utilization. Internet adoption is generally associated with different purposes, including entertainment, social interactions, information seeking, commercial activities, and education (van Boekel et al., 2017; Zheng et al., 2015). Different online activities, by their very nature, are expected to affect the well-being of older adults differently.

## 3. The Current Study

Our paper examines the impacts of online social interactions (OSIs) on the life satisfaction (LS) of older adults. Empirically, we innovate by providing a longitudinal study with panel-data methods and a cross-lagged structural equation model. These methods can, to some extent, mitigate individual endogenous bias and concerns about reverse correlation.

According to technology affordance perspectives, the various offerings of the internet can lead to different impacts [30]. Depending on the functions of the technology and the context, users’ behaviors and performances may differ (Sun et al., 2019). The existing literature has supported that diverse media usage practices can result in distinct outcomes [31]. Lifshitz, et al. [32] surveyed 306 internet users aged 50 years and above. They found that only online entertainment is associated with lower depression and higher life satisfaction, but social and information purposes are not. Szabo, Allen, Stephens and Alpass [17] found that social, instrumental, and information use online can benefit the LS of older adults in America but through different mediators.

Moreover, several computer-mediated communication theories (e.g., the Social Information Process Theory, Hyper-personal Model) suggest that the effectiveness of online communication can approach or even exceed that of face-to-face connection given sufficient interaction time [33,34]. Social media (for instance, social networking sites) have become prevalent worldwide, and a large population uses them to communicate and connect with their family, friends, and communities [35,36]. Social use of the internet provides access to more extensive social networks for older adults, enabling them to keep pace with the development of society and connect to the social world better. Such a sense of belongingness is essential for SWB [37]. He, Huang, Li, Zhou and Li [2] and Liu, et al. [38] found that Chinese older adults’ online engagement is a major predictor of their social engagement. Online engagement can enhance life satisfaction by fostering social engagement and connectedness. Networking through the internet can not only help older adults obtain social support [39,40] but also influence health-related behaviors, such as exercising, through connectedness and peer effects [41]. Other evidence suggests that social media use can also increase social capital and affect social trust [42,43].

According to the literature review and the above discussion, we hypothesize that:

**H1:** *Different online activities impact older adults’ life satisfaction differently*.

**H2:** 
*OSIs can positively affect older adults’ LS.*


**H3:** *OSIs can affect older adults’ LS by reducing loneliness, increasing social engagement, increasing pro-health behaviors, and increasing trust*.

## 4. Data and Methods

Our analysis is a secondary analysis based on the China Family Panel Studies (CFPS) data, a nationally representative, large-scale, longitudinal survey project launched in 2010 by the Institute of Social Science Survey (ISSS) of Peking University, China. The surveys adopted an implicit stratification method and multi-stage probability sampling with a population proportion base [44]. The three stratifications include county, community(village), and household (individuals) levels. It has collected data at the individual, family, and community levels and is designed to track changes in Chinese society, economy, demography, education, health, etc. We use the 2014, 2016, and 2018 waves of the CFPS to construct a panel dataset. The sample covers 29 provinces (municipalities directly under the central government and autonomous regions) in China, with a targeted sample size of 16,000 households. The empirical sample includes those who were born in 1955 and before. After matching different waves through individual unique identification numbers and keeping common variables, we constructed a balanced panel containing around 5200 older adults tracked from 2014 to 2018 (range 60–95, M = 68, SD = 5.91; 50.9% female). In the survey, life satisfaction is asked directly. All respondents answer by selecting an item ranging from 1 to 5, indicating the lowest to the highest level of satisfaction with their current lives. From 2014 to 2018, the mean value of life satisfaction increased from 3.91 to 4.26. Figure 1 provides the distributions of life satisfaction from 2014 to 2018.

Our explanatory variable of interest is OSIs. Frequencies of OSIs are surveyed directly and consistently over the three survey years. Details about OSIs are related to the usage of mobile or personal computer (PC) internet for social networking sites (e.g., Facebook, QQ space, Douban network site), microblog instant messaging (e.g., WeChat, QQ), etc. It is an ordinary variable: 0 represents no use; 1 represents once in several months; 2 represents once a month; 3 represents 2–3 times a month; 4 represents 1–2 times a week; 5 represents 3–4 times a week; and 6 represent every day. The fraction of older adults using the internet for social functions increased from 1.5% to 7.6%. The most frequent group, OSIs every day, increased from 0.7% in 2014 to 4.5% in 2018. Although social use, online study, entertainment, and commerce of older adults are lower than those for young adults, their usages are all increasing over time, and this is consistent with the increasing trend of total older internet users (see Figure A1 in Appendix A). Online education, a relatively self-oriented online activity, increased from 0.11 in 2014 to 0.18 in 2018. The least-used function for older adults is online commerce use. The most frequent adoption is using the internet for entertainment. The average value of the frequency of online entertainment, referring to online gaming, video watching, listening to music, etc., increased from 0.13 in 2014 to 0.39 in 2018. Empirically, we compare these different internet use behaviors to provide a complete picture. Because of their different values, we can conduct placebo tests and implicitly provide a robust study on the causal relationship between online networking and older adults’ LS.

In our data, we consider several channels of OSIs leading to better lives. The surveys ask how often the respondent has participated in physical activities in the past week, including public square dancing, taichi, and general sports. This decision depicts a pro-healthy behavior and reveals informal social participation offline. The average number of workouts increased from 2.7 in 2014 to 3.6 in 2018. Second, it also surveys the time allocated to watching TV and socializing with family. Third, we can also test the potential to extend the online network into a physical one through calling net-friends, meeting net-friends, or building offline friendships. Statistics and variable definitions are presented in Table 1.

Before estimation, we ran a series of Hausman tests (*χ²* = 199.61, *p* = 0.000) and Sargan–Hansen tests (*χ²* = 179.628, *p* = 0.000) to determine a random or fixed effect of the panel-data method. All test results support the individual fixed effect model and reject the fitness of random effect models. Our empirical model is set up as follows:(1)$$Life\;Satisfaction_{iot}=\alpha +\lambda_{i}+\beta_{1}OSI_{it}+\varphi X_{it}+\psi_{o}+\varepsilon_{it}$$
where *o* indicates province *o*, *i* indicates individual *i*, and *t* indicates survey year *t*. λ*_i_* represents individual fixed effects, and *ψ_o_* is the provincial fixed effects. *X_it_* is a vector of control variables, including local social status, health status, and marital status. The coefficients of OSIs, $\beta_{1}$, are expected to be positive. They reveal the effect of *OSIs* on *LS*, singling out influences of the control variables in vector *X* and individual unobserved characteristics (e.g., extraversion or introversion). The panel data we utilize contain 3 points of time, a relatively short period. During this period, social environments were stable, with no large events, policies, or economic downturns. Therefore, time indicators are not controlled for in estimations. Instead, we implement extra regressions with interactions between wave indicators and OSIs to show how the effects change over time. Meanwhile, because the older population is less likely to use the internet than the young groups, time variations of OSIs for them are relatively small. Thus, the effect we capture through Equation (1) is not expected to be very large.

For robustness checks, rather than processing OSIs as a continuous variable, we alternatively measure it with groups of categorical dummies and an indicator depicting whether they use the online social interaction function or not. Similarly, we alternatively process the other control variables in the same way. For example, health status and social status are not only controlled in terms of group dummies. Second, two-wave panel data with different time spans are used to address concerns of unobserved individual significant life events. We thirdly examine the effects across different age spans (over 60, 65, 70, 75, or 80), groups with different educational attainment as well as social status. Gender and rural–urban disparities are additionally analyzed.

## 5. Empirical Findings

### 5.1. The Effects of OSIs on the Life Satisfaction of Older Chinese Adults

The main results are presented in Table 2. LS is the dependent variable. We first run a short regression controlling individual fixed effects and then extend it by controlling local social status, health status, and marital status. Coefficients of OSIs in short regressions depict a significant positive effect on life satisfaction, 0.036. In Regression (2), the coefficient is the same after controlling for the provincial fixed effect. Regression (3) introduces controls for time-variant individual heterogeneity, including social status, health condition, and marital status. The coefficient shrinks to 0.034 but stays significantly positive. Regressions (4) and (5) use two-wave panels with different time spans. The estimates stay positively significant, and the sizes, compared with three-wave results, are larger. In Regression (6), coefficients of interaction terms reflect that the significant effect is mostly derived from the year 2018, when the OSIs become more common. These results support Hypothesis 2 and show that frequent net–society interactions extend to the offline world and create better LS for older adults in China. This is consistent with the results of Erickson and Johnson (2011) among older Canadian adults. Additionally, estimates of the time-variant variables also reveal that better local social status and health status contribute to higher LS.

### 5.2. Heterogeneity Analysis across Sub-Groups

By looking at the full sample of older adults, we may miss important differences across the distribution of older adults. The effects of OSIs may have heterogeneous impacts across different groups of older adults. Thus, we further consider different groups of older adults: males and females, rural and urban areas, different education levels, social status levels, different age spans, and geographical areas in this section. Equation (1) is re-estimated for each of these groups. Estimates of OSIs with 95% confidence intervals are reported in Figure 2, and regressions tables are presented in Appendix A. The sample is first split by age (i.e., groups at ages of 60–80, aged 65/70/75/80 and above). The pattern of coefficient sizes exhibits a strengthening trend along with aging, although the effect is not significant for the group aged 80 and above. This insignificance is probably caused by the small sample size or the cognitive difficulties of using the internet for the oldest group. Both coefficients are significantly positive in terms of gender differences. Older women seem to benefit less from OSIs (0.035 for men vs. 0.032 for women), but sizes are insignificantly different.

In contrast to the results for urban older adults, the effect of OSIs on rural older adults is larger. This suggests that OSIs are more beneficial to LS in rural locations. At the national level, the internet penetration rate is relatively lower in rural than urban areas (see Figure A2 in Appendix A), but the rural–urban digital divide has narrowed since 2019. Still, this divide is worse among the aged population, particularly for OSIs. Basic statistics in our sample show that only 4% of rural older adults report having online social interactions, while 10.2% of urban older adults report being social online. Meanwhile, the level of subjective well-being reported is relatively lower among rural compared to urban older adults. Because rural older adults are more digitally isolated and have a lower LS, our evidence suggests that more digital training and investment in digital infrastructures in rural areas can improve rural–urban equality.

Further, the sample of the older further is split by geographical areas. Regarding the regional differences, we observe that older people living in the eastern region use the internet for social interactions the most, then the northeast and central regions the least (see Figure 3). A similar pattern is found for ratios of urban older adults surveyed across regions. OSIs or internet adoption is closely related to the level of economic development. However, life satisfaction and the effects of the internet are not. The older adults living in the eastern area have the lowest life satisfaction, but this increases for those living in the western, northeastern, and central regions, respectively. As presented in the right panel of Figure 2, we found that the effects of OSIs on life satisfaction are stronger in the northeastern and central regions compared to the eastern areas. However, no evidence supports a significant effect of OSIs in western China, where older adults interact socially online the least, with only 2.4% reporting ever interacting socially online.

Finally, we analyzed the heterogeneous effects across education and social status levels. The differential impacts across different levels of educational attainment are displayed in the right panel of Figure 2. Most older adults have not completed primary schooling. This disadvantaged group has around 2304 older people, and only 4% use the internet for social interactions. This estimate suggests that they cannot benefit from OSIs. Older adults with higher education can derive more benefits from OSIs than those with senior and junior schooling, but there is not a linear pattern along with education level. Internet training interventions should focus on older adults with less education, more obstacles, and a lack of self-learning ability. The estimates of different social status groups suggest a downward linear pattern of the impacts of OSIs, and coefficients are only significant at the bottom tails. The most prominent effects are apparent for the group with the lowest social status. Unlike the findings of Hargittai and Dobransky [45] in America, OSIs contribute to narrowing the social inequality in subjective well-being across rural–urban areas and social status groups in China.

### 5.3. Explorations into the Mechanisms

To understand how OSIs enhance the LS of older adults, we further tested relationships between OSIs and channel variables. We directly examined whether OSIs generate significant impacts on life satisfaction-associated pathways through fixed-effects regressions and then evaluated the impacts of those related channels on life satisfaction. In detail, we explored whether each of the mechanisms has a direct influence on the LS of older adults. To accomplish this, we estimated the following:(2)$$Life\;Satisfaction_{iot}=\alpha +\lambda_{i}+\beta_{2}Mechanism_{it}+\varphi X_{it}+\psi_{o}+\varepsilon_{it}$$

Here, the mechanism is referred to as different channel variables. They include the number of all kinds of outdoor workouts and family dinners during the past week, smoking and reading behavior, time spent watching TV and movies, and levels of interpersonal trust. Simultaneously, we investigate whether OSIs are associated with the mechanism in question by estimating the following:(3)$$Mechanism_{it}=\alpha +\lambda_{i}+\beta_{3}OSI_{it}+\varphi X_{it}+\psi_{o}+\varepsilon_{it}$$

Both estimation results are presented in Table 3. We only report the estimators for brevity. Empirically, for a mechanism to be capable of explaining the relation between *OSI*s and SWB of older adults, estimates of both *β_2* and *β_3* are expected to be statistically significant. In other words, if the specific mechanism works, it should be a significant determinant of older adults’ life satisfaction and should be significantly influenced by *OSI*.

Internet use can generate both beneficial and detrimental effects in terms of the LS of older adults because of the diverse range of online activities. OSIs, as one specific activity online, can change time of use patterns. More time spent on OSIs likely crowds out other activities that benefit LS. For example, Moreno, et al. [46] found that internet use reduces the chance of playing sports in the young. However, estimates in the upper panel of Table 3 reveal that increased OSIs significantly increase physical activities, reduce time watching TV, and promote reading behavior in older adults. This is consistent with DiNardi et al. (2019), who found that increasing broadband usage has contributed to increased exercise in men, although they concluded that increased broadband internet access increases body weight. Additionally, this is consistent with Carrell, Hoekstra and West [41], who found that online networking can lead to more exercise through peer effects. Communications with WeChat groups of neighbors, community, or common interest groups are likely to motivate more outdoor activities with others and reduce social isolation. Another possibility is that older adults may gain health information and knowledge through OSI. For example, their friends can share health tips or health risk information through OSI. Health risks perception and benefits of physical activities increase the number of workouts, which facilitates life satisfaction.

The estimates of Regression (2) show that OSIs largely reduce time spent watching TV or movies per week and slightly increase the probability of reading paper or electronic books in the past year. Meanwhile, OSIs do not affect family dinner get-togethers, which means they do not crowd out time spent with family (see Regression (4) in Panel A). Further, evidence also shows that OSIs are not significantly related to the self-reported health condition, smoking behaviors, and increased weight. In addition, it is important to note that most older adults use mobile internet instead of computers [3], and most OSIs are fulfilled with mobile internet. Based on our evidence, OSIs are not likely to lead to being more sedentary (fewer physical activities and more screen time).

More social engagements and community activities [47,48,49,50] and an increased sense of community [51] are important channels through which internet use affects the LS of older adults. Outdoor workouts are a type of pro-healthy social participation and informal social engagement, contributing to social isolation reduction and better LS. Reading is also a determinant of happiness [52]. Because more time spent working out and reading and less time watching TV are influenced by the OSIs, the time allocation of older adults exhibits pro-healthy patterns and is beneficial to LS. In Panel B, we present estimates of these mechanisms’ variables in Equation (2). Increased physical activities and reading contribute to a higher level of life satisfaction (0.014 and 0.058). Spending more time watching TV and movies also contributes to a higher level of satisfaction (0.003).

Secondly, we consider whether OSIs truly change one’s attitudes, enlarge the social circle, and whether online friendships can be extended offline. Interpersonal trust can predict higher LS, especially for older people [53,54,55]. Influenced by the Confucian culture and moral system, Chinese citizens tend to trust those with whom they have a personal relationship (kinship or quasi-kinship) rather than strangers, in contrast to the spirit of the “stranger ethic” of modern Western philanthropy [56,57]. Our findings suggest that OSIs help change traditional attitudes toward older adults. The estimate of Regression (9) shows that online interactions significantly improved people’s trust in neighbors/strangers (0.069). Interpersonal trust is, therefore, significantly beneficial to LS in Panel B (0.023).

Estimates of Regression (5) and (7) show that online interactions increase online peer networks, the likelihood of calling net-friends (0.008), the likelihood of meeting net-friends (0.007), and the rate of becoming true friends in reality (0.006). Although the sizes are small, their positive significance suggests a nuanced potential that online connections can extend to the offline world. This finding is consistent with the statement of social network enhancement by Winstead, et al. [58]. Social network enlargement can plausibly reduce loneliness caused by physical isolation. However, in Panel B, estimates imply that the likelihood of calling and meeting net-friends, or becoming friends offline, does not directly affect the LS of older adults. One explanation is that for this population, it is rare for them to meet net-friends offline or become friends offline with those known online, unlike the younger generation. With the increase in older adults’ OSIs, though, this might work as an effective mechanism in the future.

### 5.4. Cross-Lagged Analyses of How OSIs Work through Reducing Loneliness

As discussed above, social networking online can plausibly reduce loneliness and improve the SWB of older adults [9,10,11]. Because survey questions and answer designs in the CFPS are different between 2014 and 2018, we are unable to investigate this mechanism. In this section, we further provide a cross-lagged analysis to fully understand the effects’ directions and the relationships’ causality. One advantage is that this approach allows us to provide a one-way direction link from OSIs in the past to the current LS of older adults and explore the mediating effect of loneliness reduction through the structural equation model. The basic idea is that online social interactions in 2016 are not directly related to life satisfaction in 2018. However, social interactions can work through the mediator of loneliness in 2016 and further generate a causal effect on life satisfaction in 2018. The estimation results are presented in Figure 4. As presented in Figure 4, the results show that OSIs reduce loneliness and increase life satisfaction, although there are no significant direct effects of OSIs on life satisfaction. Additionally, in this cross-sectional analysis, we did not find significant effects in terms of the number of dinners with family as well as interpersonal trust. However, the latter mediator is strongly confirmed in our panel-data methods. Our conclusions are based on the longitudinal study addressing the concerns of individual endogenous bias in cross-sectional designs.

To summarize, we conclude that OSIs generate positive effects on the LS of older adults through increasing physical activities and interpersonal trust and reducing sedentary time and loneliness. All these mechanisms are associated with informal social engagement and social isolation, and Hypothesis 3 is supported.

### 5.5. Discussion on Online Activities for Different Purposes

Not every form of online engagement significantly impacts LS or impacts it in the same way. To fully understand the effects of internet adoption, we ran regressions with different online activities, including online education, online entertainment, and online commerce. Results were presented in Table 4. As discussed earlier, studying online is mostly controlled by older adults themselves and is an online behavior that involves less connecting to other people and no social interactions. The OSIs, however, directly represent how older adults use social networking sites (e.g., Facebook, QQ space, Douban network site), microblog instant messaging (e.g., WeChat, QQ), etc. Entertainment online is using the internet for music listening, video watching, etc. Because of its entertaining nature, it can bring more joy, cultivate instant gratification, and further improve life satisfaction. The estimate of online entertainment is significantly positive, 0.028, and compared to it, the coefficient of OSIs is significantly larger. The evidence supports that OSIs are the most effective at increasing older adults’ LS. These two online activities affect the LS of older Chinese adults through different channels. Online entertainment has no significant impacts on the number of workouts, time spent watching TV or movies, and interpersonal trust, but it increases the probability of reading. Thus, online entertainment can contribute to a higher level of life satisfaction but not in the same way as OSIs.

Next, we considered the roles of online education and commerce. The regression results show that online education (e.g., searching for study materials or taking online courses, etc.) and online commerce (e.g., online shopping, busy and banking, etc.) do not significantly impact the life satisfaction of older adults. Notably, estimates suggest that all the effective mechanisms through which OSIs affect the SWB of older Chinese adults do not work in our analyses of these two online activities. An increase in online studying or commercial behaviors did not significantly increase physical activities, reduce time spent watching movies and TV, or promote trust in strangers or neighbors. Online education has been positively associated with reading behaviors, which could be a pathway to life satisfaction. Still, the overall effect of online education on life satisfaction is far from significant. Meanwhile, online commerce significantly reduces the time spent with family for dinner. Overall, different online activities impact the LS of older adults differently and through different pathways. Thus, Hypothesis 1 is supported. Additionally, the results of these two online activities can serve as placebo tests and implicitly support the causal links between OSIs and better life satisfaction of older adults.

### 5.6. Robustness with Alternative Measures

In the main analysis, we treat OSIs as a continuous variable. In Table 5, we alternatively use an indicator representing whether older adults use the internet for social interactions or not and categorical dummies of different OSIs frequencies. Empirically, we replicate previous regressions in the above tables but with these two alternative measures, respectively. Consistently, the estimates depict the same conclusion: OSIs contribute to higher-level life satisfaction. The improvement plausibly works through increasing pro-healthy behaviors (e.g., less time in front of the TV, more reading, and more physical activities) and through increasing interpersonal trust. OSIs also increase offline connections with online friends. We expect that intensive online social activity can create offline social capital, and SWB can potentially be affected in the future. In Panel B, categorical dummies for different frequency types are shown. The patterns of the coefficients imply that the effects of OSIs majorly work among intensive OISs users. The estimates of interest are significant and consistent with the main findings, suggesting that the results are not sensitive to the OSIs measurements.

## 6. Conclusions and Policy Implication

In conclusion, by using nationally representative panel data, we tested the relationship between OSIs and LS of older Chinese adults. Compared to the existing literature, we provide a longitudinal study through fixed-effects estimations and cross-lagged structural equation modeling for causal inferences. Additionally, instead of general indicators of internet use commonly used in previous studies, we adopt more specific internet use measures, providing a more detailed understanding of the effects of internet use on older adults’ life satisfaction. Our findings suggest that OSIs can improve the life satisfaction of older Chinese adults by reducing loneliness and promoting informal social engagement (joining outdoor physical activities), pro-healthy time allocation, and interpersonal trust. However, a comparison among different online activities highlights that being online is not necessarily sufficient to benefit older people’s LS. For example, online commerce and education do not raise life satisfaction levels for older adults. Overall, online social interactions are a valuable instrument for enhancing LS.

Our findings will benefit policymakers and practitioners seeking the potential for digital technologies to contribute to older adults’ subjective well-being. Since 2021, with the strong advocacy of the central government, local governments in China have continuously transformed internet applications to be suitable for older adults and promoted internet-supporting services to help older adults. Programs have been implemented to solve difficulties encountered by older adults when using intelligent technologies. These measures can help older adults overcome the digital divide and share the achievements of digital industries. Our evidence suggests that social use of the internet should be mainly considered when governments and practitioners are providing interventions to enhance the subjective well-being of older adults.

We took advantage of the nationwide representative sampling of CFPS to generalize our conclusion but were also constrained by its survey design. For example, only a direct and simple measure of life satisfaction was available. For online activities, there are no specifics for educational programs or information viewed. Thus, we only investigated the effect of frequencies of OSIs and some of the online activities on life satisfaction. Future studies could analyze the effects of specific functions and other mechanisms (e.g., formal social engagement). Moreover, we have accounted for time-invariant endogenous factors, but it is possible that time-variant factors contaminate the estimates. More advanced research designs are needed for causality, if possible—for example, the difference-in-difference approach in a longitudinal context or experimental design with a well-controlled outer environment.

## Figures and Tables

**Figure 1 healthcare-10-01964-f001:**
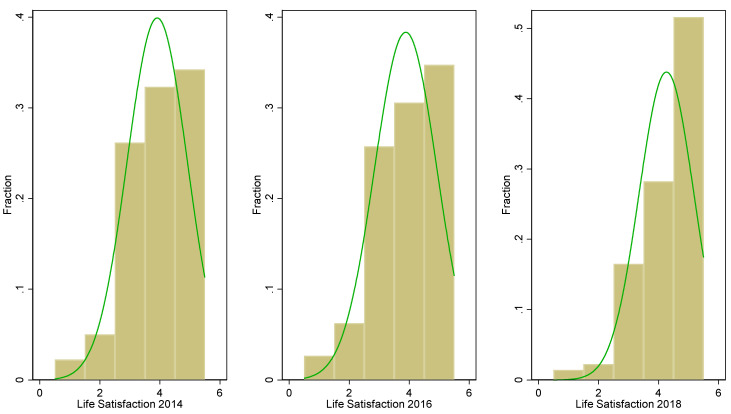
Distribution of life satisfaction 2014–2018.

**Figure 2 healthcare-10-01964-f002:**
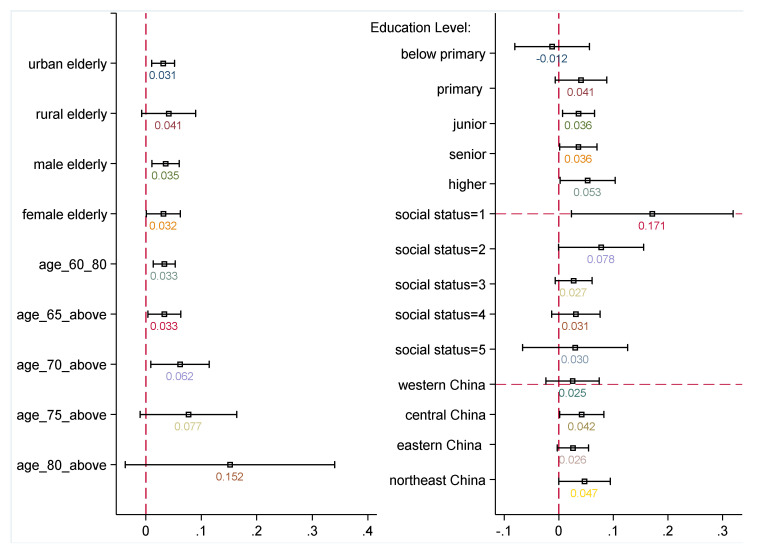
Heterogeneous effects of OSIs on older Chinese adults’ life satisfaction according to gender, residential status, age span, social status, education, and geographical area. Note: Other controls are the same as Table 2, and regressions of social status groups are not controlled for social status. Estimates of OSIs are reported in the figure with 95% confidence intervals. The regressions’ detailed results are presented in Table A1 and Table A2 in Appendix A.

**Figure 3 healthcare-10-01964-f003:**
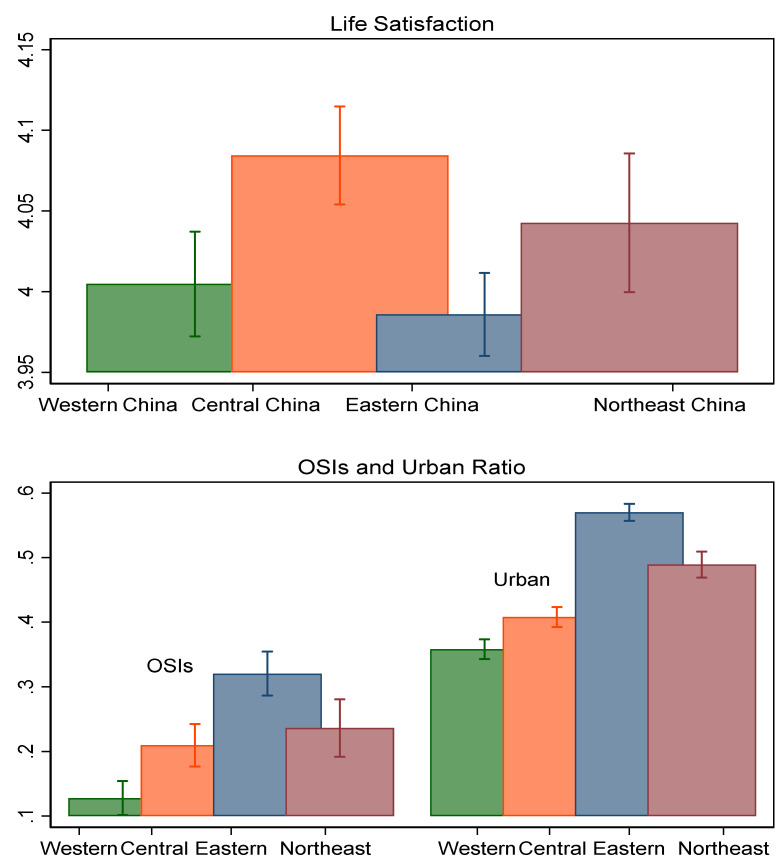
Average values of life satisfaction, OSIs, and urban proportion of older Chinese adults in different geographical areas.

**Figure 4 healthcare-10-01964-f004:**
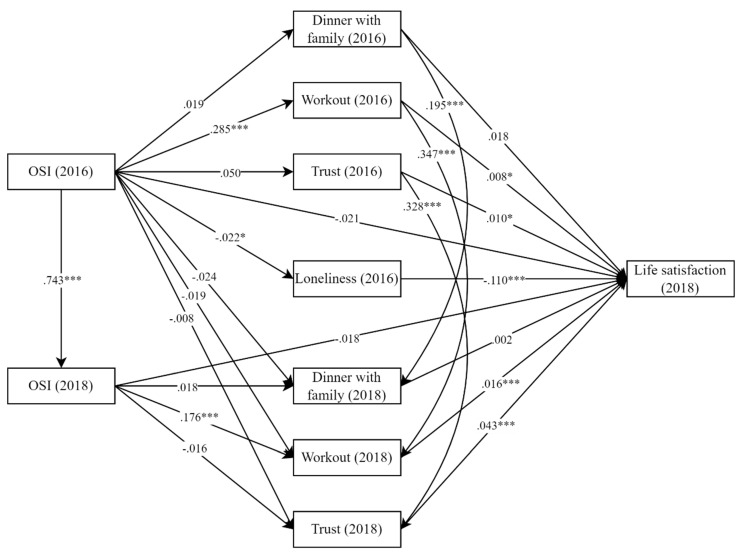
Lagged cross-mediation model with 2018 CFPS but lagged OSIs and mediators. Note: Model coefficients reported in the path diagram are standardized, and for brevity, the main paths of interest are reported, and other controls include rural/urban residential status, gender, age, and marital status. Robust model fit indices: *n* = 4637, χ² = 126.2, df = 22, CFI = 0.977, TLI = 0.916, RMSEA = 0.032, SRMR = 0.016. * *p* < 0.05, *** *p* < 0.01.

**Table 1 healthcare-10-01964-t001:** Statistics description of regression sample.

Variables	Obs.	Mean	Std. Dev.	Min	Max
Life Satisfaction: 1–5, from the lowest level of satisfaction to the highest level					
	15,386	4.022	1.000	1	5
Online Behaviors: 0 represents never; 1 represents once in several months; 2 represents once a month; 3 represents 2–3 times a month; 4 represents 1–2 times a week; 5 represents 3–4 times a week; 6 represents everyday					
Online Social Interactions	15,386	0.234	1.106	0	6
Online Education	15,386	0.137	0.830	0	6
Online Commerce	15,386	0.041	0.409	0	6
Online Entertainment	15,386	0.256	1.142	0	6
Workout: total number of all kinds of outdoor workouts during the past week					
	15,379	3.116	3.658	0	50
Time on TV and Movie: minutes per week					
	15,357	13.938	12.125	0	154
Reading: 0–1, have read paper books or electronic books or not in the past year					
	15,383	0.116	0.321	0	1
Interpersonal Trust: 0–20 sum of trust in strangers and neighbors ranging 0–10, from least to most					
	15,279	8.729	3.345	0	20
Frequency of Family Dinner Together: times during the week					
	14,480	6.568	1.533	0	7
Local Social Status:1–5, from the lowest to the highest level					
	15,386	3.220	1.119	1	5
Perceived Health Status: 1–5, from the worst to the best					
	15,386	2.498	1.211	1	5
Marital Status: 1 = single; 2 = married with spouse; 3 = cohabitation; 4 = divorced; 5 = widow					
1	15,386	0.008	0.090	0	1
2	15,386	0.822	0.383	0	1
3	15,386	0.004	0.062	0	1
4	15,386	0.010	0.099	0	1
5	15,386	0.156	0.363	0	1
Frequency of…: 0–5, from never to frequently					
Call Net-Friends	15,386	0.005	0.117	0	5
Meet Net-Friends	15,386	0.004	0.093	0	5
Become True Friends	15,386	0.005	0.104	0	5

**Table 2 healthcare-10-01964-t002:** The effects of OSIs on the LS of older Chinese adults.

Dependent Variable	Life Satisfaction							
Data and Panel Studies	(1)	(2)	(3)	(4)	(5)	(6)		
	14–18	14–18	14–18	14 & 18	16 & 18	14–18		
OSIs	0.036 ***	0.036 ***	0.034 ***	0.040 ***	0.047 ***	OSIs×2018	0.046 ***	(0.011)
	(0.010)	(0.010)	(0.010)	(0.012)	(0.014)	OSIs×2016	0.0005	(0.015)
Constant	4.014 ***	4.099 ***	3.012 ***	2.009 ***	2.865 ***	OSIs×2014	0.020	(0.025)
	(0.007)	(0.555)	(0.586)	(0.534)	(0.728)	Constant	2.844 ***	(0.240)
Controls	No	No	Yes	Yes	Yes	Controls	Yes	
Province Fixed Effect	No	Yes	Yes	Yes	Yes	Province Fixed Effect	Yes	
Individual Fixed Effect	Yes	Yes	Yes	Yes	Yes	Individual Fixed Effect	Yes	
Observations	15,499	15,493	15,386	10,030	10,118	Observations	15,386	
R-squared	0.503	0.504	0.539	0.629	0.648	R-squared	0.539	

Note: Robust standard errors are reported within paratheses. *** represent significance at 1% level. All the remaining tables keep the same format and controls. Controls include marital status, social status, and health status.

**Table 3 healthcare-10-01964-t003:** Mechanism examination of the impacts of OSIs on the LS of older adults.

	(1)	(2)	(3)	(4)	(5)	(6)	(7)	(8)	(9)
**Dependent Variables**	Workout	Smoking	Time watching TV and movies	Reading	Number of dinners with family	Calls with net-friends	Meetings with net-friends	Became friends in reality	Interpersonal trust
**OSIs**	**0.095 *****	**0.000**	−0.318 ***	0.006 **	0.007	0.008 ***	0.007 ***	0.006 ***	0.069 **
	**(0.035)**	**(0.002)**	(0.109)	(0.003)	(0.016)	(0.001)	(0.001)	(0.001)	(0.033)
**Observations**	**15,382**	**15,389**	15,359	15,386	14,380	15,389	15,389	15,389	15,274
**R-squared**	**0.557**	**0.874**	0.618	0.638	0.496	0.549	0.505	0.526	0.550
**Dependent Variable**		Life Satisfaction							
Mechanism of Interest	Workout	Smoking	Time watching TV and movies	Reading	Number of dinners with family	Calls with net-friends	Meetings with net-friends	Became friends in reality	Interpersonal trust
	0.014 ***	0.043	0.003 ***	0.058 *	0.004	−0.024	0.006	−0.016	0.023 ***
	(0.003)	(0.042)	(0.001)	(0.035)	(0.006)	(0.086)	(0.103)	(0.094)	(0.003)
Observations	15,379	15,386	15,356	15,383	14,379	15,386	15,386	15,386	15,272
R-squared	0.540	0.538	0.539	0.538	0.538	0.538	0.538	0.538	0.542

Note: Robust standard errors are reported within parentheses. *** represent significance at 1% level; ** represent significance at 5% level; * represent significance at 10% level. All the remaining tables keep the same format and controls.

**Table 4 healthcare-10-01964-t004:** Different online activities and the LS of older adults.

	(1)	(2)	(3)	(4)	(5)	(6)
**Dependent Variables**	LS	Workouts	Time watching TV and movies	Dinners with family	Reading	Interpersonal trust
**Online Education**	0.016	0.053	−0.095	0.034	0.024 ***	0.041
	(0.014)	(0.050)	(0.153)	(0.023)	(0.004)	(0.046)
Constant	3.184 ***	2.274 ***	14.314 ***	6.726 ***	0.099 ***	7.610 ***
	(0.036)	(0.129)	(0.399)	(0.060)	(0.010)	(0.120)
Observations	15,386	15,382	15,359	14,380	15,386	15,274
R-squared	0.538	0.557	0.617	0.496	0.639	0.550
**Dependent Variables**	LS	Workouts	Time watching TV and movies	Dinners with family	Reading	Interpersonal trust
**Online Commerce**	0.021	0.021	−0.027	−0.085 **	−0.006	0.066
	(0.024)	(0.085)	(0.262)	(0.039)	(0.007)	(0.079)
Constant	3.185 ***	2.280 ***	14.303 ***	6.735 ***	0.103 ***	7.612 ***
	(0.036)	(0.129)	(0.398)	(0.060)	(0.010)	(0.120)
Observations	15,386	15,382	15,359	14,380	15,386	15,274
R-squared	0.538	0.557	0.617	0.496	0.638	0.550
**Dependent Variables**	LS	Workouts	Time watching TV and movies	Dinners with family	Reading	Interpersonal trust
**Online Entertainment**	0.028 ***	0.028	−0.115	0.010	0.006 **	0.018
	(0.010)	(0.036)	(0.112)	(0.016)	(0.003)	(0.034)
Constant	3.179 ***	2.274 ***	14.334 ***	6.728 ***	0.101 ***	7.611 ***
	(0.036)	(0.129)	(0.399)	(0.060)	(0.010)	(0.121)
Observations	15,386	15,382	15,359	14,380	15,386	15,274
R-squared	0.539	0.557	0.618	0.496	0.638	0.550

Note: Robust standard errors are reported within paratheses. *** represent significance at 1% level; ** represent significance at 5% level. All the remaining tables keep the same format and controls. Controls include marital status, social status, and health status.

**Table 5 healthcare-10-01964-t005:** Alternative measures for robustness checks.

Dependent Variables	LS	Workout	Time Watching TV and Movies	Weekly Family Dinner Together	Calling net-Friends	Meeting net-Friends	Becoming Friends in Reality	Reading	Interpersonal Trust
**Panel A**	(1)	(2)	(3)	(4)	(5)	(6)	(7)	(8)	(9)
Whether OSIs	0.187 ***	0.473 **	−1.733 ***	0.017	0.051 ***	0.046 ***	0.046 ***	0.035 **	0.325 *
	(0.052)	(0.187)	(0.574)	(0.085)	(0.006)	(0.005)	(0.005)	(0.015)	(0.173)
Constant	3.178 ***	2.260 ***	14.380 ***	6.729 ***	−0.002	−0.002	0.000	0.101 ***	7.601 ***
	(0.036)	(0.129)	(0.399)	(0.060)	(0.004)	(0.003)	(0.004)	(0.010)	(0.120)
Observations	15,386	15,382	15,359	14,380	15,389	15,389	15,389	15,386	15,274
R-squared	0.539	0.557	0.618	0.496	0.550	0.506	0.528	0.638	0.550
**Panel B**	(1)	(2)	(3)	(4)	(5)	(6)	(7)	(8)	(9)
Base Group:	Never Use								
1. Once in several months	0.276	0.435	−0.858	−0.052	0.040	0.039 *	0.039	0.204 ***	0.531
	(0.238)	(0.854)	(2.630)	(0.400)	(0.028)	(0.023)	(0.025)	(0.068)	(0.788)
2. Once a month	0.253	0.206	−2.397	−0.007	0.038	0.038	0.036	−0.043	0.200
	(0.242)	(0.869)	(2.678)	(0.397)	(0.028)	(0.023)	(0.025)	(0.069)	(0.803)
3. 2–3 times a month	0.076	0.713	−1.620	0.172	0.027	0.027	0.114 ***	0.168 ***	−0.434
	(0.178)	(0.638)	(1.966)	(0.312)	(0.021)	(0.017)	(0.019)	(0.051)	(0.592)
4. 1–2 times a week	0.168	−0.035	−1.948 *	−0.078	0.094 ***	0.097 ***	0.100 ***	−0.021	0.225
	(0.104)	(0.373)	(1.151)	(0.171)	(0.012)	(0.010)	(0.011)	(0.030)	(0.345)
5. 3–4 times a week	0.164	0.457	−1.116	−0.185	0.105 ***	0.078 ***	0.081 ***	−0.051 *	0.132
	(0.109)	(0.391)	(1.205)	(0.177)	(0.013)	(0.010)	(0.011)	(0.031)	(0.361)
6. OSIs everyday	0.204 ***	0.624 ***	−1.870 ***	0.096	0.026 ***	0.025 ***	0.012 *	0.058 ***	0.498 **
	(0.064)	(0.230)	(0.708)	(0.105)	(0.007)	(0.006)	(0.007)	(0.018)	(0.213)
Observations	15,386	15,382	15,359	14,380	15,389	15,389	15,389	15,386	15,274
R-squared	0.539	0.557	0.618	0.496	0.553	0.509	0.532	0.639	0.550

Note: Robust standard errors are reported within paratheses. *** represent significance at 1% level; ** represent significance at 5% level; * represent significance at 10% level. All the remaining tables keep the same format and controls. Controls include marital status, social status, and health status.

## Data Availability

We confirm that the data will be available when required and that we obtained permission to reproduce material from other sources. We will make our data available in an open repository as required.

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
