# Peer review of "The Effects of Online Social Interactions on Life Satisfaction of Older Chinese Adults: New Insights Based on a Longitudinal Approach"

_healthcare, 2022, doi:10.3390/healthcare10101964_

Round 1
Reviewer 1 Report
Authors have done an interesting study on The Effects of Different Online Functions on the Elderly’ Well- being: Evidence from a Longitudinal Study.
Introduction has well written but authors have started discussing about their study in the introduction section, I would recommend to limit the introduction section to make a case for current study and discuss previous studies on the topic and refrain from discussing their study results in the discussion. Otherwise introduction contains good information and discuss some of the bigger studies that have been published.
Methods are meticulous, described in detail and are statistically sound with multiple model to remove bias. There can be some simplified explanation about the methods for the readers.
Results are well described with the help of figures and tables which I truly appreciate. It is interesting to notice some of the results are counterintuitive such as OSI increasing physical activity but the authors have provided good explanation regarding the plausible causes, I truly appreciate their mediation model, which is very helpful and informative. It was further interesting to see that online educational activities were not much beneficial to improve SWB.
Discussion and conclusion are sound and data driven, it would be interesting to see if authors can follow this cohort in the future and perform similar analysis a few years later to show change in trend of OSI affecting SWB in the similar manner or there will be changes as more people who qualify to be in the elder cohort might have more educational level and thus affecting their internet use.
Minor spellings such as in line 177 needs correction
Author Response
Dear Editor and Referees,
We would like to express our appreciation for the revision opportunity. We are grateful to you and the referees for the suggestions and comments. Please find enclosed the revised manuscript along with a letter of response to the referees' comments.
- Introduction has well written but authors have started discussing about their study in the introduction section, I would recommend to limit the introduction section to make a case for current study and discuss previous studies on the topic and refrain from discussing their study results in the discussion. Otherwise, introduction contains good information and discuss some of the bigger studies that have been published.
Response: In the revised manuscript, we have removed the results of our study in the introduction section and limited the introduction section to motivations, previous studies, and contributions description.
- Methods are meticulous, described in detail and are statistically sound with multiple models to remove bias. There can be some simplified explanation about the methods for the readers.
Response: Thank you very much! In the current manuscript, we have tried to revise our expressions, and use simplified explanations. Also, we have added interpretations of the interested coefficients in the models.
- Results are well described with the help of figures and tables which I truly appreciate. It is interesting to notice some of the results are counterintuitive such as OSI increasing physical activity but the authors have provided good explanation regarding the plausible causes, I truly appreciate their mediation model, which is very helpful and informative. It was further interesting to see that online educational activities were not much beneficial to improve SWB.
Response: In the future, we would further investigate the effects of online education on other perspectives of the lives of the elderly.
- Discussion and conclusion are sound and data driven, it would be interesting to see if authors can follow this cohort in the future and perform similar analysis a few years later to show change in trend of OSI affecting SWB in the similar manner or there will be changes as more people who qualify to be in the elder cohort might have more educational level and thus affecting their internet use.
Response: Appreciate this insightful comment. We indeed carefully think about generalization specialties and common impacts among different generations. Given the current population, there is a significant digital divide between the younger and older generations. In the future, today’s youth will be elderly. The elderly of the future is better educated and familiar with new media. The proportion of OSIs would be larger and they comprehend more internet functions than the cohorts we studied. Can similar effects be observed for the elderly of the future? Or new determinations can be found. We expect that technology will keep changing, which may create new communication dynamics between individuals. In such circumstances, the effects of “technological” social interaction may vary. Those would be very interesting research questions. With more longitudinal data, we would pursue this direction.
- Minor spellings such as in line 177 needs correction
Response: Thank you very much for pointing this out. The paper has been formatted and I am not sure which line is exactly referred to. But we have carefully gone over it in detail and done proofreading. Spelling errors are revised, e.g., “loneliness”.
Reviewer 2 Report
This is an interesting study examining the effect of online social interactions on subjective well-being in older adults. The paper is well-written and I believe it may contribute to the literature well. I have several comments to improve the manuscript further:
1. In the introduction, I feel that it is quite misleading for the authors to mention that most of the studies are cross-sectional. There are many recent studies on computer use and internet use that are longitudinal in nature. These studies should be highlighted in the Introduction.
Relevant recent papers that should be discussed and incorporated:
Cognitive, social, emotional, and subjective health benefits of computer use in adults: A 9-year longitudinal study from the Midlife in the United States (MIDUS). (2020). Computers in Human Behavior, 104, 106179. Longitudinal analysis of the relationship between purposes of internet use and well-being among older adults. (2019). The Gerontologist, 59(1), 58-68.
2. It will be useful for the authors to provide more information on the Chinese Family Panel Study. What is the sampling method of the study? What is the procedure? As it is mentioned that the sample is a nationally representative sample, there is a need for more justification in this issue
3. The missing data and data imputation technique should be highlighted in the method section Ibrahim, J. G., & Molenberghs, G. (2009). Missing data methods in longitudinal studies: A review. Test, 18(1), 1-43.
4. More information should be provided regarding the measures used to assess online social interactions and well-being. The validity and reliability of the measures were not discussed in the manuscript
5. "Our evidence suggests that social use of the internet can focus on
providing those interventions." The final sentence in the conclusion is awkward and can be improved further. The authors should also consider to tone down their discussion due to the use of correlational instead of experimental design.
6. It will be important for the authors to highlight some of the limitations of the study in the discussion.
Author Response
Dear Editor and Referees,
We would like to express our appreciation for the revision opportunity. We are grateful to you and the referees for the suggestions and comments. Please find enclosed the revised manuscript along with a letter of response to the referees' comments.
This is an interesting study examining the effect of online social interactions on subjective well-being in older adults. The paper is well-written and I believe it may contribute to the literature well. I have several comments to improve the manuscript further:
- In the introduction, I feel that it is quite misleading for the authors to mention that most of the studies are cross-sectional. There are many recent studies on computer use and internet use that are longitudinal in nature. These studies should be highlighted in the Introduction.
Relevant recent papers that should be discussed and incorporated:
- Cognitive, social, emotional, and subjective health benefits of computer use in adults: A 9-year longitudinal study from the Midlife in the United States (MIDUS). (2020). Computers in Human Behavior, 104, 106179.
- Longitudinal analysis of the relationship between purposes of internet use and well-being among older adults. (2019). The Gerontologist, 59(1), 58-68.
Response: Thank you for sharing the recent papers. We have updated and discussed these references in the introduction section of the revision.
“One close study is from Szabo et al. (2019) in which they evaluated the influences of different activities online of respondents in 2013 on their well-being in 2016. Similarly, Hartanto et al. (2020) employed a two-wave cross-lagged design to evaluate the relationship between computer use and healthy, cognitive, and social benefits among middle-aged and older people. Both studies focus on the American elderly. Strictly speaking, their longitudinal mediation analyses are cross-sectional studies with lagged Structural Equation Modelling. Our study improves the majority of existing literature by providing a longitudinal study in China with individual fixed-effect models. In this panel data estimation method, we eliminate endogenous selection bias arising from individual as well as other time-invariant unobserved factors. Second, previous research focused on the frequency of internet use or a general indicator for internet utilization rather than engagement in different online uses. Our study also complements the existing research by making comparisons among different online behaviors, instead of a general indicator, and investigating potential causal channels.”
- It will be useful for the authors to provide more information on the Chinese Family Panel Study. What is the sampling method of the study? What is the procedure? As it is mentioned that the sample is a nationally representative sample, there is a need for more justification in this issue
Response: Our analysis is based on the data from the China Family Panel Studies (CFPS), which is a nationally representative, large-scale, longitudinal survey project launched in 2010 by the Institute of Social Science Survey (ISSS) of Peking University, China. The surveys adopted an implicit stratification method and multi-stage probability sampling with a population proportion base (Xie & Hu, 2014). The three stratifications include county, community(village), and household (individuals) levels. It has collected data at the individual, family, and community levels and is designed to track changes in Chinese society, economy, demography, education, health, etc. The sample covers 29 provinces (municipalities directly under the central government and autonomous regions) in China with a targeted sample size of 16,000 households.
In the revision, we have added the above description into the data section.
Xie, Y., & Hu, J. (2014). An introduction to the China family panel studies (CFPS). Chinese sociological review, 47(1), 3-29. DOI: 10.2753/CSA2162-0555470101.2014.11082908
- The missing data and data imputation technique should be highlighted in the method section Ibrahim, J. G., & Molenberghs, G. (2009). Missing data methods in longitudinal studies: A review. Test, 18(1), 1-43.
Response: We appreciate your thoughtful suggestion. Attrition is a common issue in longitudinal experimental studies. Our analysis is based on the data from the China Family Panel Studies (CFPS), which is a nationally representative, large-scale, longitudinal survey project. The survey targeted a sample of 16000 households originally. But in each wave, the number of surveyed individuals has increased. One reason is household expanded because of new-born or marriage. Another reason is new families included. Using the 2014, 2016 and 2018 wave, we matched them by individual unique identification number and then constructed a balanced panel, containing around 5200 older adults tracked from 2014 to 201he 8 (Range 60–95, M = 68, SD =5.91; 50.9% female). Among this sample, the rate of missing observation (missing value of some variables) is trivial, less than 1%. Our main results are barely affected. Accordingly, the missing data in our sample should be categorized as missing completed at random (MCAR). In the likelihood and Bayesian paradigm, and when mild regularity conditions are satisfied, the MCAR mechanisms are ignorable, in the sense that inferences can proceed by analyzing the observed data only, without explicitly addressing a (parametric) form of the missing data mechanism.
The variable of Frequency of Family Dinner Together is the one with more missing observation (14480, compared with the total 15386). For robustness, we have replaced the missing observations with mean values of at its belonged birth cohort, education level group, social status group, and region. Consistent results are found. Moreover, we have generated a dummy indicating the missing observations and estimated its relationships to variables of online activities and other control variables as suggested. No systematic patterns are found.
- More information should be provided regarding the measures used to assess online social interactions and well-being. The validity and reliability of the measures were not discussed in the manuscript.
Response: We took advantage of the nationwide representative sampling of CFPS to generalize our conclusion, but were also constrained by its survey design. For example, only a direct and simple measure for well-being, life satisfaction ranging from 1-5, was available. This measure is commonly used in the existing literature. All respondents answer by selecting an item ranging from 1 to 5, indicating the lowest to the highest level of satisfaction with their current lives (See lines 186-190).
Also, Frequencies of OSIs were used. It is an ordinary variable: 0 represents never use; 1 represents once in several months; 2, once a month; 3 represents 2-3 times a month; 4 represents 1-2 times a week; 5 represents 3-4 times a week;6 represent every day (see 193-200). Therefore, Cronbach’s ? as well as other tests are not applicable in this scenario. We understand this constraint and we have added a discussion of the limitations of our study in the conclusion section. Detailed statistics and variable definitions are presented in Table 1.
- "Our evidence suggests that social use of the internet can focus on
providing those interventions." The final sentence in the conclusion is awkward and can be improved further. The authors should also consider to tone down their discussion due to the use of correlational instead of experimental design.
Response: Thank you so much for this suggestion. We really appreciate it. In the revision, we revised it to be “Our evidence suggests that social use of the internet should be mainly considered when governments and practitioners are providing interventions to enhance the well-being of the elderly”.
Meanwhile, we also pay attention to the expressions avoiding strong statements related to causality. We believe that we did move a step forward and it is not a simple correlational analysis with the cross-sectional design. In the future, we plan to adopt advanced research designs if possible (e.g., Difference-in-Difference estimation and experimental intervention).
- It will be important for the authors to highlight some of the limitations of the study in the discussion.
Response: In the revision, we add a short paragraph to discuss the limitations of the current study and future studies in the conclusion section. Please see page 16.
“We took advantage of the nationwide representative sampling of CFPS to generalize our conclusion but were also constrained by its survey design. For example, only a direct and simple measure for life satisfaction was available. For online activities, there are no specifics for educational programs or information viewed. Thus, we only investigate the effect of frequencies of OSIs and some of the online activities on life satisfaction. Future studies could analyze the effects of specific functions as well as other mechanisms (e.g., formal social engagement). Moreover, we have accounted for time-invariant endogenous factors but it is possible that time-variant factors will contaminate the estimates. More advanced research designs are needed for causality, if possible, for example, difference in difference approach in longitudinal context or experimental design with a well-controlled outer environment.”
Reviewer 3 Report
In this article, the authors describe research on an important topic such as different online functions on the elderlies. Modern society is increasingly using IOT tools and applications, and this cannot but affect the change in the perception of people of different categories. This point applies to various areas such as online commerce, online education and online entertainment. However, the older generation has difficulty adapting to modern online services.
The article made a detailed literature analysis of existing studies in this area, proposed its own statistical analysis, and made the results of the increase in the interest of older people in the field of online services, applicable to analyze the situation in different regions of China.
However, authors should pay attention to the following comments:
1.The main remark is the non-obvious relation of the content of the article to the general theme of the journal. Could you make this more obvious?
2.The research is based on data from 2014-2018. Is there more up-to-date data?
3. In the article, you define the circle of subjects who were born after 1955 (line 181). Has an earlier generation been analyzed? However, below you provide a sample for the age category (60-95). This question needs to be clarify
4.Formulas 1, 2-1 go beyond the rules for the design of the article. There are also no comments and interpretation of the coefficients of formula 1 (alpha, beta, etc.)
5.What is the main metric for assessing life satisfaction? Is there any scale for verifying your data?
Author Response
Dear Editor and Referees,
We would like to express our appreciation for the revision opportunity. We are grateful to you and the referees for the suggestions and comments. Please find enclosed the revised manuscript along with a letter of response to the referees' comments.
Comments:
In this article, the authors describe research on an important topic such as different online functions on the elderlies. Modern society is increasingly using IOT tools and applications, and this cannot but affect the change in the perception of people of different categories. This point applies to various areas such as online commerce, online education and online entertainment. However, the older generation has difficulty adapting to modern online services.
The article made a detailed literature analysis of existing studies in this area, proposed its own statistical analysis, and made the results of the increase in the interest of older people in the field of online services, applicable to analyze the situation in different regions of China.
However, authors should pay attention to the following comments:
- The main remark is the non-obvious relation of the content of the article to the general theme of the journal. Could you make this more obvious?
Response: Isolation poses a significant risk to health and well-being. This paper looks into how the use of information technology can reduce isolation, contribute to social connectedness, and then help the elderly reduce loneliness and enjoy a better later life. This study relates directly to social connectedness and mental health. Thus, we believe it relates to the general theme of the journal and fits the special issue of "Social Connectedness and Isolation in Relation to Health, Well-Being, Resiliency, Illness, and Recovery".
- The research is based on data from 2014-2018. Is there more up-to-date data?
Response: The latest wave of CFPS is 2020 which was conducted after the outbreak of COVID-19. Environments changed a lot and most of the work moved online. We have worked on the dataset to explore the impacts of COVID-19 on the lives of the elderly. However, this current paper is related to the impact of OSIs and other functions of internet on the older adults. For the generalization and validity of our conclusions, we think it had better use waves before the pandemic and get a less contaminated result.
- In the article, you define the circle of subjects who were born after 1955 (line 181). Has an earlier generation been analyzed? However, below you provide a sample for the age category (60-95). This question needs to be clarified
Response: We apologize for the typo. It should be those who were born in 1955 and before. We have revised this in the current paper.
- Formulas 1, 2-1 go beyond the rules for the design of the article. There are also no comments and interpretation of the coefficients of formula 1 (alpha, beta, etc.)
Response: Formulas 1 and 2-1 are two commonly used model specifications for fixed effect models in the longitudinal study. We estimated equations 1, 2-1, and 2-2 to capture the influence of OSI on SWB and the potential mechanisms. In the current version, we have added interpretations for the main and interested coefficients, but not all of them due to limited space. We also try our best to provide simplified explanations in the method part (see lines 130-135 and page 10).
- What is the main metric for assessing life satisfaction? Is there any scale for verifying your data?
Response: We took advantage of the nationwide representative sampling of CFPS to generalize our conclusion, but were also constrained by its survey design. For example, only a direct and simple measure for life satisfaction ranging from 1-5 was available. This measure is commonly used in the existing literature.
Round 2
Reviewer 2 Report
The authors have addressed all my comments well. I appreciate all their efforts. Well done!
Author Response
Dear Referee,
Thank you very much for your supports. In the recent minor revision, we have re-done the spell check and proofread. We believe the language has imporved.
Reviewer 3 Report
all comments have been corrected, the text of the article has been improved
Author Response
Dear Referee,
Thank you for your supports.
During this current revision, we have our manuscript checked by a
colleague fluent in English writing (we think he is) . Also, we have checked for ourselves for three rounds. We believe that the quality has been improved.